# DIFFUSION PRIORS FOR BAYESIAN 3D RECONSTRUCTION FROM INCOMPLETE MEASUREMENTS

## ABSTRACT

Many inverse problems are ill-posed and need to be complemented by prior information that restricts the class of admissible models. Bayesian approaches encode this information as prior distributions that impose generic properties on the model such as sparsity, non-negativity or smoothness. However, in case of complex structured models such as images, graphs or three-dimensional (3D) objects, generic prior distributions tend to favor models that differ largely from those observed in the real world. Here we explore the use of diffusion models as priors that are combined with experimental data within a Bayesian framework. We use 3D point clouds to represent 3D objects such as household items or biomolecular complexes formed from proteins and nucleic acids. We train diffusion models that generate coarse-grained 3D structures at a medium resolution and integrate these with incomplete and noisy experimental data. To demonstrate the power of our approach, we focus on the reconstruction of biomolecular assemblies from cryo-electron microscopy (cryo-EM) images, which is an important inverse problem in structural biology. We find that posterior sampling with diffusion model priors allows for 3D reconstruction from very sparse, low-resolution and partial observations.

## 1 INTRODUCTION

Inverse problems are encountered in many different scientific fields. The basic setting is that we observe noisy and incomplete data $\boldsymbol{y}$ and seek to find a model $\boldsymbol{x}$ that predicts mock data via a forward model $\mathcal{A}$ such that $\boldsymbol{y} \approx \mathcal{A}(\boldsymbol{x})$. An important subclass are linear models where $\mathcal{A}$ is a linear operator. Well-known inverse problems are deconvolution or tomography.

The challenge in solving inverse problems stems from the fact that they tend to be ill-posed meaning that many models can produce highly similar data and/or the reconstructed model can be very sensitive to noise. The remedy is to combine a reconstruction loss with a regularizer. Well-studied regularizers are Tikhonov regularization (aka ridge regression), sparsity, and non-negativity.

Bayesian inference offers a powerful framework to tackle inverse problems. The conditional probability $p(\boldsymbol{y} \,|\, \boldsymbol{x})$, the likelihood, relates the data $\boldsymbol{y}$ to the mock data $\mathcal{A}(\boldsymbol{x})$ via a noise model. A common assumption is independent Gaussian noise resulting in the likelihood

$$p(\boldsymbol{y} \,|\, \boldsymbol{x}) \propto \exp\left(-\frac{\|\boldsymbol{y} - \mathcal{A}(\boldsymbol{x})\|^2}{2\sigma^2}\right) . \tag{1}$$

Maximizing the likelihood is then equivalent to standard least-squares fitting.

The prior probability $p(\boldsymbol{x})$ encodes data-independent knowledge about a particular model $\boldsymbol{x}$; its negative logarithm $-\log p(\boldsymbol{x})$ can be viewed as a regularizer. The posterior of the model is

$$p(\boldsymbol{x} \,|\, \boldsymbol{y}) = \frac{p(\boldsymbol{y} \,|\, \boldsymbol{x}) \, p(\boldsymbol{x})}{p(\boldsymbol{y})} \tag{2}$$

with model evidence $p(\boldsymbol{y}) = \int p(\boldsymbol{y} \,|\, \boldsymbol{x}) \, p(\boldsymbol{x}) \, d\boldsymbol{x}$. In case of a Gaussian likelihood, maximization of $\log p(\boldsymbol{x} \,|\, \boldsymbol{y})$ is equivalent to regularized least-squares fitting.

Often detailed knowledge about reasonable solutions is available but difficult to capture by the standard priors that are typically used to tackle inverse problems. For example, cryo-electron microscopy

(cryo-EM) aims to reconstruct the three-dimensional structure of macromolecular complexes from two-dimensional (2D) projections. Cryo-EM images are typically very noisy with signal-to-noise ratios (SNR) far below one. On the other hand, a large body of knowledge has been accumulated over the past six decades, including hundreds of thousands of experimentally determined biomolecular structures that are stored in the Protein Data Bank (PDB) (Berman et al., 2000). Experimentally determined structures exhibit recurrent features such as alpha-helices and beta-strands and preferences for the proximity and packing of amino acids and entire subunits. This detailed information is not captured by standard priors used in cryo-EM reconstruction packages such as cryoSPARC (Punjani et al., 2017) or RELION (Scheres, 2012). These approaches represent the structure as voxel grid and use generic priors enforcing non-negativity or penalizing high-frequency contributions. If one were to sample volumes from the corresponding prior, the sampled structures would not resemble any of the known biomolecular structures. Here we try to encode the rich knowledge available in the PDB as a diffusion model prior. We test 3D reconstruction from sparse, low-resolution and partial measurements by posterior sampling with diffusion models as priors.

## 1.1 CONTRIBUTION

Our contributions are as follows: We propose a method to reconstruct 3D structures from 2D projections that utilizes diffusion models as priors. Using diffusion priors has previously not been explored to solve the 3D reconstruction problem in cryo-EM. The combination of the diffusion model prior with a likelihood allows us to reconstruct 3D structures from very sparse observations such as 2D projections, low-resolution structures and known structures of subunits. This is achieved with diffusion-based posterior sampling (DPS) (Chung et al., 2023) a method that has not yet been investigated in the context of 3D data. We combine DPS with optimized diffusion schedules and second-order correction steps with adaptable noise injection (Karras et al., 2022) to improve sample quality and runtime. We demonstrate the fidelity and flexibility of our method on highly complex and diverse datasets of 3D point clouds from ShapeNet and the PDB.

We emphasize that the reconstruction problem which we solve differs from the problem of reconstructing a 3D surface from a 2D surface color image, which is tackled by, for example, Point-E (Nichol et al., 2022), Shape-E (Jun & Nichol, 2023), PC[2] (Melas-Kyriazi et al., 2023), One-2-3-45 (Liu et al., 2023) and BDM (Xu et al., 2024). The main difference is that in our work the 2D observations are projections that provide information about the density across the full volume rather than only information about the surface. In addition, our approach is also capable of conditioning on coarse-grained or partial observations of the 3D structure. Moreover, the reconstruction problem in cryo-EM aims to reconstruct the internal structure not only the surface.

## 2 BACKGROUND ON DIFFUSION MODELS

Diffusion models have gained wide recognition in the field of generative modeling (Sohl-Dickstein et al., 2015; Song & Ermon, 2019; Ho et al., 2020; Song et al., 2021), particularly in image synthesis, where diffusion models have demonstrated their capability by surpassing former leading models in key metrics (Dhariwal & Nichol, 2021) and continue to set new records (Karras et al., 2024). In generative modeling, the main goal is to learn a sampler for an unknown distribution $p_0$ from i.i.d. samples $\boldsymbol{x}(0)_i \sim p_0$ that serve as training data. A diffusion model tries to achieve this goal by approximating a probability flow from a latent Gaussian distribution $p_T$ to the unknown target $p_0$.

For this purpose, a **forward process** from the target distribution $p_0$ to the latent distribution $p_T$ is defined in terms of a stochastic differential equations (SDE) of the form

$$d\boldsymbol{x} = \boldsymbol{f}(\boldsymbol{x}, t)\, dt + g(t)\, d\boldsymbol{w}_t, \tag{3}$$

where $\boldsymbol{w}_t$ is a Wiener process, $\boldsymbol{f}(\cdot, t) : \mathbb{R}^d \to \mathbb{R}^d$ is the *drift* of $\boldsymbol{x}(t)$ and $g : \mathbb{R} \to \mathbb{R}$ is the *diffusion coefficient* (Song et al., 2021). Starting at time $t = 0$ with samples $\boldsymbol{x}(0) \sim p_0$ from the target distribution, process (3) is designed such that it gradually destroys the information content of the samples $\boldsymbol{x}(0)$ by transforming them into samples $\boldsymbol{x}(T)$ from an isotropic Gaussian.

A diffusion model aims to represent the **reverse process** from $p_T$ to $p_0$ such that we can draw noise from a Gaussian distribution and slowly transform it into samples from the data distribution $p_0$.

Anderson (1982) showed that the forward process (3) has a reverse process of the form

$$dx = \left[ f(x, t) - g(t)^2 \nabla_x \log p_t(x) \right] dt + g(t) \, dw_t \tag{4}$$

with $p_t(x(t)) = \int p_{0t}(x(t) \mid x(0)) \, p_0(x(0)) \, dx(0)$ being the marginal distribution of $x(t)$ where $p_{0t}$ is the perturbation kernel from time 0 to $t$. The score $\nabla_x \log p_t(x)$ of the marginals is unknown and has to be approximated with a parametric *score model* $s_{\theta}(x(t), t)$.

**Diffusion model training** works by applying gradient descent to the *denoising score matching* (DSM) objective to train $s_{\theta}$:

$$\min_{\theta} \mathbb{E}_{t, x(0), x(t)} \left[ \lambda(t) \left|\left| \nabla_{x(t)} \log p_{0t}(x(t) \mid x(0)) - s_{\theta}(x(t), t) \right|\right|^2 \right] \tag{5}$$

where $t \sim p_{\text{train}}$, $x(t) \sim p_t$, $x(0) \sim p_0$ and $x(t) \sim p_{0t}(\cdot \mid x(0))$ with the loss weighting $\lambda : \mathbb{R}^+ \to \mathbb{R}^+$. In DSM, we only need to evaluate the score of the perturbation kernel $p_{0t}$, which is easy to calculate for suitable choices of the drift and diffusion coefficient (consider, for instance, the *variance exploding* or *variance preserving* schedules in Song et al. (2021)). More background on the training process can be found in Appendix A.2. After training, the score model $s_{\theta}$ can be used as a replacement for $\nabla_x \log p_t(x)$ to **generate new data** by sampling the latent model $p_T$ and simulating the reverse SDE in equation (4) backward in time. The reverse SDE can be simulated with numerical methods such as Euler-Maruyama, starting from $T$ and ending shortly before 0 to avoid numerical errors.

## 2.1 DIFFUSION MODELS IN 3D

Apart from the 2D image domain, diffusion models have been employed to estimate the distribution of 3D objects. Various representations have been used including point clouds (Luo & Hu, 2021; Vahdat et al., 2022; Nichol et al., 2022; Zhou et al., 2021), meshes and implicit neural representations (Jain et al., 2021; Jun & Nichol, 2023; Erkoç et al., 2023) such as neural radiance fields (Mildenhall et al., 2021). Here, we employ a point cloud representation and adopt the point transformer architecture from Nichol et al. (2022). This representation allows us to model 3D volume densities such as cryo-EM maps efficiently, unlike meshes, which only model the surface. Furthermore, the point cloud representation simplifies the process of developing likelihoods for the cryo-EM reconstruction problem. In addition, we avoid any kind of latent diffusion (as, for example, proposed by Vahdat et al. (2022)) for which likelihood guidance is more difficult (Song et al., 2024).

## 2.2 DIFFUSION POSTERIOR SAMPLING

In many practical applications such as text-to-image or class-to-image generation, the focus is on sampling from the posterior $x(0) \sim p_0(\cdot \mid y)$ given some input $y$. In this case, our unconditional score $\nabla_{x(t)} \log p_t(x(t))$ will be extended to

$$\nabla_{x(t)} \log p_t(x(t) \mid y) \overset{\text{Bayes rule}}{=} \nabla_{x(t)} \log p_t(y \mid x(t)) + \nabla_{x(t)} \log p_t(x(t)). \tag{6}$$

Given pairs of training data $\{(x(0)_i, y_i)\}$, we could train a diffusion prior plus a classifier $p_t(y \mid x(t))$ and use its score $\nabla_{x(t)} \log p_t(y \mid x(t))$ during inference for **classifier guidance** (Dhariwal & Nichol, 2021). Another popular option is to perform **classifier-free guidance** and directly train $\nabla_{x(t)} \log p_t(x(t) \mid y)$ (Ho & Salimans, 2022). For example, Zhou et al. (2021) used this approach for 3D shape completion and 3D shape reconstruction from a single depth map.

Another line of work attempts to **avoid task-specific training** and instead uses the known forward model to guide the generation process (Chung et al., 2023; 2022; Ho et al., 2022; Lugmayr et al., 2022; Song et al., 2021; Trippe et al., 2023a;b; Dou & Song, 2024; Cardoso et al., 2023). In tasks with known forward model like inpainting, shape completion or colorization, we have access to a likelihood $p_0(y \mid x(0))$ based on the noiseless data $x(0)$. Chung et al. (2023) make use of this likelihood by approximating the score of the posterior by

$$\nabla_{x(t)} \log p_t(x(t) \mid y) \approx \zeta \nabla_{x(t)} \log p_0(y \mid D_{\theta}(x(t), t)) + \nabla_{x(t)} \log p_t(x(t)) \tag{7}$$

with weighting $\zeta > 0$ and *denoising function* $D_{\theta}$, which is an estimator of $D(x(t), t) := \mathbb{E}_{x(0) \sim p(\cdot \mid x(t))}[x(0)]$ that is learnt during the training of the diffusion model (see Section 3.1 and

Appendix A.2). This approximation approach, called **reconstruction guidance** in Ho et al. (2022), has been applied across multiple contexts with prominent results in ill-posed inverse problems of 2D images (Chung et al., 2023; 2022; Ho et al., 2022; Trippe et al., 2023a). Simpler approaches such as the **replacement method** of Song et al. (2021) are computationally cheaper because they do not need additional backpropagation. However, the replacement method sometimes suffers from severe artifacts (Lugmayr et al., 2022; Chung et al., 2023). Most recently, several approaches used reweighing schemes within the **Sequential Monte Carlo** (SMC) framework to derive exact methods for diffusion posterior sampling (Trippe et al., 2023a;b; Cardoso et al., 2023; Dou & Song, 2024). However, the guarantee of exactness is not of practical relevance in our case, because the required number of particles in SMC tends to be excessively large (Gupta et al., 2024).

## 3 THEORETICAL FRAMEWORK

Our theoretical framework is inspired by existing diffusion models and uses reconstruction guidance provided by forward models for 3D reconstruction from sparse observations in 2D and 3D.

### 3.1 3D DIFFUSION PRIOR TRAINING AND SAMPLING

We follow the design choice recommendations of Karras et al. (2022) using $\boldsymbol{f}(\boldsymbol{x}, t) = \mathbf{0}$ and $g(t) = \sqrt{2t}$ which yield the forward diffusion SDE $d\boldsymbol{x} = \sqrt{2t}\, d\boldsymbol{w}_t$ and the perturbation kernel $p_{0t}(\boldsymbol{x}(t) \,|\, \boldsymbol{x}(0)) = \mathcal{N}(\boldsymbol{x}(t); \boldsymbol{x}(0), t^2 \boldsymbol{I})$. For the loss weighting $\lambda(t)$, $p_{\text{train}}(t)$ and the score model parameterization $\boldsymbol{s}_{\boldsymbol{\theta}}(\boldsymbol{x}(t), t) = (\boldsymbol{D}_{\boldsymbol{\theta}}(\boldsymbol{x}(t), t) - \boldsymbol{x}(t))/t^2$ we also followed Karras et al. (2022) (more details can be found in the Appendix A.2).

During inference time, we utilize the more general version of the reverse SDE presented by Karras et al. (2022) which has the same marginals as $d\boldsymbol{x} = \sqrt{2t}\, d\boldsymbol{w}_t$ and gives us more flexibility in choosing favorable sampling schemes:

$$d\boldsymbol{x} = -[t + \beta(t)t^2]\nabla_{\boldsymbol{x}} \log p_t(\boldsymbol{x})dt + \sqrt{2\beta(t)t^2}\, d\boldsymbol{w}_t \tag{8}$$

where $\beta : \mathbb{R}^+ \to \mathbb{R}^+$ is a function that controls how noisy the trajectory behaves. The choice $\beta(t) = 1/t$ results in Eq. (4) as a special case, whereas $\beta(t) = 0$ yields an ordinary differential equation (ODE) called the *flowODE*. In practice, the score of the marginals $\nabla_{\boldsymbol{x}} \log p_t(\boldsymbol{x})$ is replaced by the score estimator $(\boldsymbol{D}_{\boldsymbol{\theta}}(\boldsymbol{x}, t) - \boldsymbol{x})/t^2$ and the differential equation must be solved backward in time by a numerical integrator such as Euler-Maruyama for a specific time interval $t \in [t_{\min}, t_{\max}]$ where $t_{\min} > 0$. The time interval must be discretized into $N$ time steps $\{t_{\max} = t_0 > \ldots > t_{N-1} = t_{\min} > t_N = 0\}$. More time steps result in a more accurate simulation of the SDE, but also increase the number of network function evaluations (NFE). Accurate simulation of the SDE can be especially difficult in areas with a high curvature in the trajectory, which is typically prominent at smaller $t$. We therefore adopt the time step heuristic of Karras et al. (2022): $t_i = (t_{\max}^{1/\rho} + \frac{i}{N-1}(t_{\min}^{1/\rho} - t_{\max}^{1/\rho}))^\rho$ with $i < N$, $\rho \geq 1$ and $t_N = 0$ where an increase in $\rho$ leads to more time steps in the lower part of the time frame. We found that $\rho = 3$ works well for sampling 3D point clouds. Algorithm 1 with $\nabla \log \tilde{p}_t(\boldsymbol{x} \,|\, \boldsymbol{y}) = (\boldsymbol{D}_{\boldsymbol{\theta}}(\boldsymbol{x}, t) - \boldsymbol{x})/t^2$ implements unguided diffusion prior sampling using Euler-Maruyama with correction step.

### 3.2 DIFFUSION POSTERIOR SAMPLING FOR 3D RECONSTRUCTION

To sample the trained diffusion prior in the light of observations $\boldsymbol{y}$ originating from a known forward process, we use reconstruction guidance (Chung et al., 2023). In contrast to Chung et al. (2023), we apply a more advanced diffusion schedule (EDM (Karras et al., 2022) instead of VP-SDE (Song et al., 2021)) to enhance the capabilities of the proposed guidance strategy. We supplement the schedule with a stochastic Euler-Maruyama integrator that uses a second-order correction step, because both the use of stochasticity (solving an SDE rather than an ODE) and second-order samplers have been shown to improve image generation performance in the unconditional setting (Karras et al., 2022). We observed that this also holds for our conditional setting in 3D (see Table 2). For conditional generation, we extend the score of the marginals from the diffusion prior $\nabla_{\boldsymbol{x}(t)} \log p_t(\boldsymbol{x}(t))$ with an approximate score of the perturbed likelihoods:

$$\nabla_{\boldsymbol{x}(t)} \log p_t(\boldsymbol{x}(t)) + \zeta \nabla_{\boldsymbol{x}(t)} \log p_0(\boldsymbol{y} \,|\, \boldsymbol{D}_{\boldsymbol{\theta}}(\boldsymbol{x}(t), t)) =: \nabla_{\boldsymbol{x}(t)} \log \tilde{p}_t(\boldsymbol{x}(t) \,|\, \boldsymbol{y}) \tag{9}$$

with $\zeta = \alpha(t)/\sqrt{\log p_0(\boldsymbol{y} \mid \boldsymbol{D_\theta}(\boldsymbol{x}(t),t))}$ following Chung et al. (2023). Algorithm 1 illustrates our method for conditional generation with reconstruction guidance. In order to apply this methodology to reconstruct partially observed 3D volumes represented as point clouds, we now list the subsequent forward processes.

**Single 2D projection to 3D.** In the simplest version of the reconstruction problem, we partially observe a single 2D projection of a 3D object in a known orientation. Here we represent the structure of an object as a 3D point cloud $\boldsymbol{x}(0) \in \mathbb{R}^{N \times 3}$ with $N$ points and the corresponding 2D projection $\boldsymbol{y}_1 \in \mathbb{R}^{M \times 2}$ as a 2D point cloud consisting of $M$ points. We define the likelihood of observing the projection $\boldsymbol{y}_1$ given $\boldsymbol{x}(0)$ as $p_0(\boldsymbol{y}_1 \mid \boldsymbol{x}(0)) \propto \exp\left(-E_1(\boldsymbol{x}(0))\right)$ where the energy is defined as

$$E_k(\boldsymbol{x}(0)) := \min_{\boldsymbol{P} \in \mathcal{P}^{N \times N}} \left|\left| \boldsymbol{P}\boldsymbol{U}\boldsymbol{y}_k - (\boldsymbol{x}(0)\boldsymbol{R}_k)_{:,(1,2)} \right|\right|_{\mathrm{F}}^2 \tag{10}$$

for $k = 1$ with permutation matrices $\mathcal{P}^{N \times N} \subset \{0,1\}^{N \times N}$, orthogonal matrix $\boldsymbol{R}_k \in O(3)$, Frobenius norm $||\cdot||_{\mathrm{F}}$ and the linear operator $\boldsymbol{U} \in \{0,1\}^{N \times M}$ that upsamples[1] $\boldsymbol{y}_k$ by randomly redrawing points. The permutation matrix $\boldsymbol{P}$ assigns each point in $\boldsymbol{U}\boldsymbol{y}_k$ to a single point in the rotated and projected object $(\boldsymbol{x}(0)\boldsymbol{R}_k)_{:,(1,2)}$. The introduction of $\boldsymbol{P}$ arises from the assumption of a hidden one-to-one correspondence between the upsampled points $\boldsymbol{U}\boldsymbol{y}_k$ and the points in $\boldsymbol{x}(0)$. The inner optimization problem is a *linear assignment problem* (Crouse, 2016) that can be solved exactly in polynomial time by using the *Hungarian method* (Kuhn, 1955). We stress that due to the missing correspondence information, the 3D reconstruction problem from 2D projections with known orientations is non-trivial and severely ill-posed.

**Multiple 2D projections to 3D.** To generalize the above forward process, we consider the case of observing $K$ projections $\boldsymbol{y} = \{\boldsymbol{y}_1, \ldots, \boldsymbol{y}_K\}$ of an object $\boldsymbol{x}(0)$ from known orientations $\boldsymbol{R} = \{\boldsymbol{R}_1, \ldots, \boldsymbol{R}_K\}$. Then the likelihood of observing the set of projections $\boldsymbol{y}$ from orientations $\boldsymbol{R}$ given $\boldsymbol{x}(0)$ is the product of all independent observations:

$$p_0(\boldsymbol{y} \mid \boldsymbol{x}(0)) = \prod_k p(\boldsymbol{y}_k \mid \boldsymbol{x}(0)) \propto \exp\left( - \sum_k E_k(\boldsymbol{x}(0)) \right) \tag{11}$$

**Coarse to fine grained.** We can also guide the diffusion prior by a 3D point cloud with fewer points $M < N$ representing a low-resolution version $\boldsymbol{y}_{\mathrm{cg}} \in \mathbb{R}^{M \times 3}$ of $\boldsymbol{x}(0) \in \mathbb{R}^{N \times 3}$. From this coarser observation, we want to infer a higher resolution structure. In order to characterize the relation between different resolutions, we employ a likelihood similar to the one used for 2D projections, $p_0(\boldsymbol{y}_{\mathrm{cg}} \mid \boldsymbol{x}(0)) \propto \exp\left(-E_*(\boldsymbol{x}(0))\right)$ where the energy is defined as

$$E_{\mathrm{cg}}(\boldsymbol{x}(0)) := \min_{\boldsymbol{P} \in \mathcal{P}^{N \times N}} ||\boldsymbol{P}\boldsymbol{U}\boldsymbol{y}_{\mathrm{cg}} - \boldsymbol{x}(0)||_{\mathrm{F}}^2. \tag{12}$$

**From subunit to full 3D reconstruction.** If available we can further update our prior knowledge encoded in the diffusion model by utilizing information about parts or subunits of the unknown 3D structure. Thus we define the energy for the likelihood $p_0(\boldsymbol{y}_{\mathrm{su}} \mid \boldsymbol{x}) \propto \exp(-E_{\mathrm{su}}(\boldsymbol{x}(0)))$ of observing the subunit $\boldsymbol{y}_{\mathrm{su}} \in \mathbb{R}^{L \times 3}$ given $\boldsymbol{x}(0) \in \mathbb{R}^{N \times 3}$ by

$$E_{\mathrm{su}}(\boldsymbol{x}(0)) := \min_{\boldsymbol{P} \in \mathcal{P}^{L \times N}} ||\boldsymbol{P}\boldsymbol{x}(0) - \boldsymbol{y}_{\mathrm{su}}||_{\mathrm{F}}^2 \tag{13}$$

with partial permutation matrices $\mathcal{P}^{L \times N} \subset \{0,1\}^{L \times N}$ that pick $L$ out of $N$ points in $\boldsymbol{x}(0)$ to create a one-to-one correspondence to the $L$ points in $\boldsymbol{y}_{\mathrm{su}}$.

We can also combine likelihoods for all possible observations $\boldsymbol{y} = \{\boldsymbol{y}_{\mathrm{su}}, \boldsymbol{y}_{\mathrm{cg}}, \boldsymbol{y}_1, \ldots, \boldsymbol{y}_K\}$ of the 3D structure to update the prior knowledge encoded in the diffusion prior. To enable the assignment of importance or uncertainty to each dataset, we can weight the corresponding energies:

$$p_0(\boldsymbol{y} \mid \boldsymbol{x}(0)) \propto \exp\left( -w_{\mathrm{su}}E_{\mathrm{su}}(\boldsymbol{x}(0)) - w_{\mathrm{cg}}E_{\mathrm{cg}}(\boldsymbol{x}(0)) - \sum_k w_k E_k(\boldsymbol{x}(0)) \right) \tag{14}$$

with weights $w_{\mathrm{su}}, w_{\mathrm{cg}}, w_k \geq 0$, coarse-grained structure $\boldsymbol{y}_{\mathrm{cg}}$, subunit $\boldsymbol{y}_{\mathrm{su}}$, 2D observations $\{\boldsymbol{y}_1, \ldots, \boldsymbol{y}_K\}$, orientations $\boldsymbol{R}$ and 3D structure $\boldsymbol{x}(0)$. In the experiments of this work, we apply equal weighting of $1/|\boldsymbol{y}|$ to all the observations. The likelihood guidance of the diffusion prior allows us to flexibly incorporate all this information with varying shapes, thereby avoiding task-specific retraining.

---

[1] Here we look at the case $M \leq N$, however this formulation can also be used to downsample $\boldsymbol{y}_k$ if $M > N$.

---

**Algorithm 1** Approximate posterior sampling with correction step

---

1: **Input:** Noise control function $\beta$, time steps $\{t_0 > t_1 > \ldots > t_N = 0\}$, observations $\boldsymbol{y}$

2: **Output:** Approximate sample from $p_0(\boldsymbol{x}(0) \,|\, \boldsymbol{y})$

3: $\boldsymbol{x}(t_0) \sim \mathcal{N}(\boldsymbol{0}, t_0^2 \mathbf{I})$

4: **for** $i \in \{0, \ldots, N-1\}$ **do**

5:      $\Delta t \leftarrow (t_i - t_{i+1})$

6:      $\boldsymbol{x}(t_{i+1}) \leftarrow \boldsymbol{x}(t_i) + t_i \nabla \log \tilde{p}_t(\boldsymbol{x}(t_i) \,|\, \boldsymbol{y}) \Delta t$

7:      **if** $t_{i+1} \neq 0$ **then**                       $\triangleright$ correction step + noise injection

8:          $\boldsymbol{d} \leftarrow (t_i + \beta(t_i) t_i^2) \left[ \nabla \log \tilde{p}_t(\boldsymbol{x}(t_i) \,|\, \boldsymbol{y}) + \nabla \log \tilde{p}_t(\boldsymbol{x}(t_{i+1}) \,|\, \boldsymbol{y}) \right] \Delta t / 2$

9:          $\boldsymbol{n} \sim \mathcal{N}\left(\boldsymbol{0}, 2\beta(t_i) t_i^2 \Delta t \mathbf{I}\right)$

10:        $\boldsymbol{x}(t_{i+1}) \leftarrow \boldsymbol{x}(t_i) + \boldsymbol{d} + \boldsymbol{n}$

11:      **end if**

12: **end for**

13: **return** $\boldsymbol{x}$

---

## 4 EXPERIMENTS

To demonstrate the fidelity and flexibility of our approach, we conducted multiple experiments. For this, we performed training on multiple different 3D datasets and tested their usefulness on a variety of 3D reconstruction tasks.

### 4.1 DIFFUSION PRIOR TRAINING

We trained diffusion priors for each of the three datasets from multiple domains each differing in their level of complexity.

**(A) ShapeNet-Chair**: 2658 point clouds from the training split of the ShapeNet dataset in the category "Chair" accessed via PyTorch Geometric (Chang et al., 2015; Fey & Lenssen, 2019). During training, we randomly subsampled 1024 points from each point cloud and applied random orthogonal transformations to augment the dataset.

**(B) ShapeNet-Mixed**: 10693 point clouds from the training split of the ShapeNet dataset in the categories "Airplane", "Bag", "Cap", "Car", "Chair", "Guitar", "Laptop", "Motorbike", "Mug", "Pistol", "Rocket", "Skateboard" and "Table" (all categories from ShapeNet with point clouds larger than or equal to 1024) accessed via PyTorch Geometric (Chang et al., 2015; Fey & Lenssen, 2019). Again, we applied subsampling and augmentation with random orthogonal transformations to the training data.

**(C) CryoStruct**: 6629 point clouds representing mixture models of size 1024 constructed from the 3D atom positions of biomolecular complexes from the PDB contained in the train split of the curated Cryo2StructDataset (Giri et al., 2024). The mixture models were created using the scikit-learn GaussianMixture method with covariance matrix shared among the components (Pedregosa et al., 2011). We also augmented the dataset by randomly rotating the biomolecular complexes.

The point clouds in all three datasets are centered and scaled so as to fit into the $[-1, 1]$ cube. Figures 3, 4, and 5 in the appendix present images of unconditional samples from the diffusion priors. Following the methodology of Yang et al. (2019), we present the 1-nearest neighbor accuracy (1-NNA), coverage (COV), and minimum matching distance (MMD) in Table 3 in the appendix to quantify the performance of the diffusion model.

### 4.2 3D RECONSTRUCTION ON SHAPENET

To demonstrate the performance and flexibility of our method on the widely used ShapeNet benchmark (Chang et al., 2015), we conducted experiments across nine different configurations. An advantage of the ShapeNet reconstruction tasks is that it is easier to visually judge the quality of the

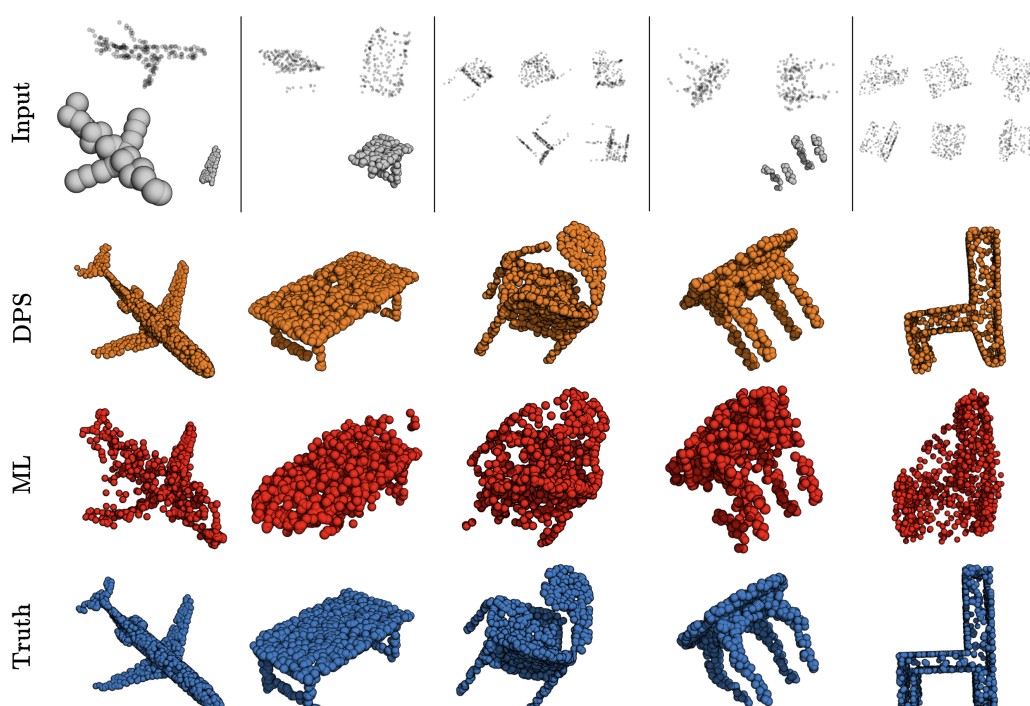

Figure 1: Results for five different reconstruction tasks. In all examples, the ML reconstruction has a higher likelihood of observing the input data than the models obtained with approximate DPS. However, the ML-based models show a higher reconstruction error than those from DPS. The results are also part of the tests presented in Table 1 and correspond to rows 9, 8, 1, 8 and 2 (from left to right).

reconstructions than for the CryoStruct reconstruction tasks. In each setting, we took the first 100 instances from both the ShapeNet-Chair and ShapeNet-Mixed test set as ground truth and created sparse observations $y$. These observations include 2D projections, coarse-grained point clouds, or subunits. The 2D projections are constructed by sampling points from the ground truth and applying a random orthogonal transformation to the sub-sampled points before projecting them onto the $xy$-plane. The coarse-grained point clouds are constructed by taking the means from a mixture model fitted to the ground truth point cloud. A subunit, i.e. a partial structure, corresponds to a single $k$-means cluster selected randomly from the ground truth.

We applied our version of approximate DPS (see Algorithm 1) to generate ten 3D reconstructions per instance using only 40 time steps (additional details on the parameters can be found in Section A.3 of the Appendix). We compared our method to the ML approach obtained by maximizing the same log-likelihood that was also used to guide the diffusion prior during approximate DPS. Starting from 10 different random clouds with points uniformly distributed in $[-1, 1]^3$, we performed gradient descent for 100 steps using the Adam optimizer with a learning rate of 0.01 (Kingma & Ba, 2014). By using the same likelihoods without the diffusion model, we can assess how much we gain in 3D reconstruction performance by utilizing a diffusion prior. Similar to the approach of Yang et al. (2019), we measure the 3D reconstruction error between a reconstructed point cloud and the ground truth with the Chamfer Distance (CD) and the Earth Movers Distance (EMD). The values in Table 1 are the means and standard deviations of all $100 \times 10$ reconstruction errors measured in CD and EMD as well as the negative log-likelihood (energy $E$) of the corresponding forward model.

Table 1 shows that, as expected, in most cases the maximum likelihood approach creates 3D reconstructions with a higher likelihood (lower energy $E$) of observing the input data $y$ than DPS. However, in the face of the ill-posedness of the reconstruction tasks, it is not sufficient to simply optimize the likelihood. This explains why the incorporation of the diffusion prior consistently results in better reconstruction errors in all test cases for both EMD and CD, although for most test

Table 1: Results from the 3D reconstruction task from sparse data. Tests were conducted on the test partition of the ShapeNet (Mixed, Chair) datasets under various configurations, altering the number of points per projection, coarse-grained structure and subunit. We compared our variant of approximate diffusion posterior sampling (DPS) to the maximum likelihood (ML) approach. To quantify the error between the reconstructions and the ground truth point clouds we calculated the mean Chamfer Distance (CD) and mean Earth Movers Distance (EMD) over in total 1k reconstructions (10 samples for each of the 100 test instances). For further analysis we also show the energy of the forward model ($E$).

| ShapeNet category | Method | Projection points | Number of projections | Coarse grained points | Subunit points | CD($[\times 10^2], \downarrow$) | EMD($[\times 10^2], \downarrow$) | $E([\times 10^3], \downarrow)$ |
|---|---|---|---|---|---|---|---|---|
| Chair | DPS | 200 | 5 | - | - | **9.98 ± 2.38** | **8.32 ± 1.77** | 3.80 ± 1.02 |
|  | ML |  |  |  |  | 13.30 ± 2.00 | 11.03 ± 1.69 | **3.68 ± 0.88** |
| Chair | DPS | 200 | 6 | - | - | **9.71 ± 1.81** | **7.78 ± 1.35** | 3.98 ± 0.87 |
|  | ML |  |  |  |  | 12.53 ± 1.53 | 10.04 ± 1.30 | **3.87 ± 0.79** |
| Mixed | DPS | 400 | 4 | - | - | **10.56 ± 4.09** | **8.18 ± 3.40** | 2.41 ± 1.01 |
|  | ML |  |  |  |  | 12.37 ± 2.34 | 10.21 ± 2.09 | **2.38 ± 0.79** |
| Mixed | DPS | 400 | 5 | - | - | **9.29 ± 2.65** | **7.00 ± 2.22** | **2.30 ± 0.89** |
|  | ML |  |  |  |  | 11.78 ± 1.88 | 9.39 ± 1.77 | 2.72 ± 1.27 |
| Chair | DPS | 300 | 1 | 30 | - | **10.38 ± 2.24** | **10.66 ± 3.23** | 6.21 ± 1.76 |
|  | ML |  |  |  |  | 12.40 ± 2.16 | 11.89 ± 2.84 | **5.04 ± 1.52** |
| Mixed | DPS | 300 | 1 | 30 | - | **9.36 ± 2.23** | **9.21 ± 2.47** | 5.57 ± 2.11 |
|  | ML |  |  |  |  | 11.99 ± 1.89 | 11.08 ± 2.01 | **5.13 ± 1.69** |
| Chair | DPS | 200 | 2 | - | ≈ 256 | **13.21 ± 5.69** | **12.86 ± 5.62** | 2.51 ± 0.66 |
|  | ML |  |  |  |  | 16.98 ± 4.58 | 16.63 ± 4.85 | **1.84 ± 0.63** |
| Mixed | DPS | 200 | 2 | - | ≈ 256 | **11.11 ± 4.13** | **11.19 ± 5.09** | 2.47 ± 0.86 |
|  | ML |  |  |  |  | 18.14 ± 6.28 | 17.99 ± 6.59 | **2.22 ± 1.05** |
| Mixed | DPS | 200 | 1 | 30 | ≈ 128 | **8.55 ± 1.97** | **9.38 ± 1.96** | 4.17 ± 1.64 |
|  | ML |  |  |  |  | 11.19 ± 1.82 | 10.90 ± 1.88 | **3.80 ± 1.49** |

Table 2: Evaluation of the improvement we obtain by switching from integrating the *flowODE* ($\beta(t) = 0$) using Euler's method in A to the integration of the SDE ($\beta(t) = 1/t$ if $t > 0.15$ and else 0) using the Euler–Maruyama method in B. In C, we observe that the reconstruction error is lowered further by adding a second-order correction step. The test errors have been studied on the ShapeNet-Mixed reconstruction task given a subunit with ≈ 256 points and two projection images with 200 points each (row 8 in Table 1). In all three schedules, we used 79 NFE which accounts to 79 time steps in A and B and 40 time steps in C.

|  |  | CD($[\times 10^2], \downarrow$) | EMD($[\times 10^2], \downarrow$) | $E([\times 10^3], \downarrow)$ |
|---|---|---|---|---|
| A | Euler ODE | 14.38 ± 5.64 | 13.98 ± 5.94 | 3.09 ± 1.19 |
| B | + noise | 11.80 ± 3.94 | 11.71 ± 5.03 | 2.61 ± 0.85 |
| C | + correction step | **11.11 ± 4.13** | **11.19 ± 5.09** | **2.47 ± 0.86** |

cases the likelihoods obtained with DPS are worse than those obtained with ML. Prominent example reconstructions that demonstrate the superior performance of DPS are shown in Figure 1. The diffusion prior helps navigate the space of possible 3D reconstructions with high likelihood toward those with typical ShapeNet structures, information that is not sufficiently provided by the observations $y$ themselves. The structural models obtained with DPS are also visually much closer to the ground truth and show a lower degree of heterogeneity

## 4.3 DIFFUSION POSTERIOR SAMPLING FOR CRYO-EM

We also benchmark posterior sampling with diffusion priors on various reconstruction tasks arising in the context of cryo-EM reconstruction. We are mostly interested in sparse data scenarios. This might appear to be at odds with the fact that cryo-EM tends to produce many hundreds of thousands of images. Our interest is in reconstructing intermediate resolution structures from very few images, with the goal of elucidating structural differences between individual copies of the biomolecule. These structural variations are expected to occur, because biomolecular complexes are flexible and undergo conformational changes. Conformational heterogeneity is often linked to the biological

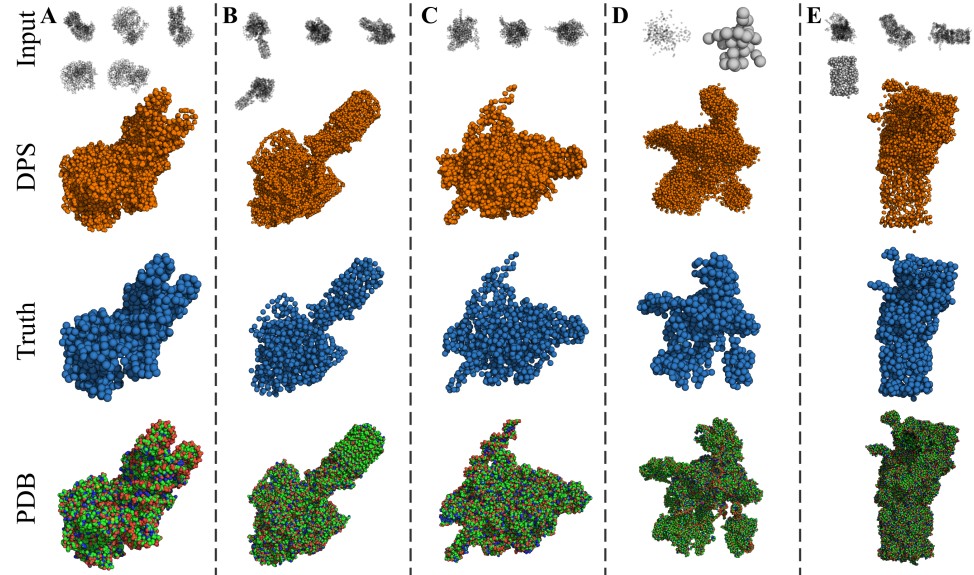

Figure 2: Outcomes for five cryo-EM reconstruction tasks. The top row shows the sparse input measurements. The second row shows all ten point clouds generated with DPS. The third row shows the 1024 component means of a mixture model fitted to the atomic models (last row). **(A)** Nucleosome-CHD4 from five projections (PDB code 6ryr). **(B)** F-ATP Synthase from four projections (PDB code 6rdm). **(C)** RNA polymerase transcription open promoter complex with Sorangicin from three projections (PDB code 6vvy). **(D)** Human spliceosome after Prp43 loaded from one projection and a low-resolution structure consisting of 40 particles (PDB code 6id1). **(E)** 26S proteasome from three projections and a known 20S structure (PDB code 6fvt).

function of a macromolecular complex and of particular interest to the structural biologist (Toader et al., 2023).

We designed various benchmarks based on a held-out set of 100 structures from Cryo2StructDataset that were not used in the training of the diffusion prior. The reconstruction tasks involve sparse 2D and/or 3D information. Again, as a baseline we used ML models obtained by maximizing the likelihood without the diffusion prior (a detailed presentation of the results can be found in the Supplementary Material, Sections A.4.1 to A.4.7). We generated ten models with and without diffusion model per reconstruction task. To assess the accuracy of the model structures, we compare them against the atomic structure deposited in the PDB and the point cloud obtained by fitting a 1024-component mixture of Gaussians used for the generation of the input measurements. The 3D points generated by ML and DPS tend to concentrate in $[-1, 1]^3$. Before a meaningful comparison between the ground truth and model structures can be made, we first need to scale the model points so as to match the physical units of the coordinates in the PDB file (which are in Å). We achieve this by matching the radii of gyration. However, there could still be a mismatch between the ground truth and the scaled model resulting from a relative rotation and translation (rigid transformation) between the two point clouds. We estimate the optimal alignment of both point clouds by maximizing the kernel correlation (Tsin & Kanade, 2004).

After scaling and superposition, we can meaningfully compare model point clouds against the atomic and coarse-grained ground truth structures. We assess the accuracy of the models with the root mean square deviation (RMSD) which is commonly used to compare biomolecular structures. Since there is no one-to-one correspondence between the points in the cloud representing the ground truth (all heavy atoms in the PDB file or component means of the Gaussian mixture) and the models computed with ML or DPS, we compute RMSD $= (\frac{1}{N} \sum_{n=1}^{N} \|\boldsymbol{x}_n - \boldsymbol{x}'_{\ell_n}\|^2)^{1/2}$ where $\ell_n \in \{1, \dots, 1024\}$ encodes the correspondence between points $\boldsymbol{x}_n$ representing the ground truth and points $\boldsymbol{x}'_m$ representing the model (where $m \in \{1, \dots, 1024\}$). In case the ground truth is represented by all heavy atoms, we set $\ell_n = \mathrm{argmin} \|\boldsymbol{x}_n - \boldsymbol{x}'_m\|$ (where "argmin" runs over all $m$) and

$N$ is the total number of heavy atoms in the PDB file. The 100 PDB structures in the test set vary largely in the number of heavy atoms from $N = 2178$ to $N = 110541$. In case the ground truth is represented by the 1024 component means of the Gaussian mixture (also referred to as "subsampled structure" in the following), we compute $\ell_n$ by solving the linear assignment problem that matches the 1024 points representing the ground truth against the 1024 in the model (in this case $N = 1024$).

Figure 2 shows representative cryo-EM reconstructions for five different sparse data scenarios. Figure 2A shows the results for a nucleosome-CHD4 complex (PDB code 6ryr, 17820 heavy atoms). Five 2D projections served as input for DPS reconstruction. The RMSD between the ten DPS models and the ground truth is $3.56 \pm 0.04$ Å (atomic structure) and $2.05 \pm 0.09$ Å (subsampled structure). We also inferred the structure of F-ATP synthase (PDB code 6rdm, 33891 heavy atoms) from 4 projections (Fig. 2B). The RMSDs between the DPS models and the ground truth is $4.46 \pm 0.02$ Å (atomic structure) and $2.83 \pm 0.03$ Å (subsampled structure). RNA polymerase transcription open promoter complex with Sorangicin (PDB code 6vvy, 30033 heavy atoms) was inferred from three projections (Fig. 2C). The RMSDs between the DPS models and the ground truth is $4.87 \pm 0.77$ Å (atomic structure) and $3.39 \pm 1.21$ Å (subsampled structure). These tests show that intermediate resolution structures can be computed from very few 2D projections.

A common scenario in cryo-EM is that a low-resolution structure is already known and the goal of a cryo-EM study is to furnish structural details at higher resolution. This scenario was tested on the human intron lariat spliceosome (PDB code 6id1, 79882 heavy atoms). The structural models were computed from a single projection and a low-resolution structure represented by only 40 points (Fig. 2D). The RMSD between DPS models and the ground truth is $10.35 \pm 0.20$ Å (atomic structure) and $9.82 \pm 0.15$ Å (1024 component means). Because the structure is huge and the input data for DPS are very sparse, the RMSD is worse than in the previous examples. Nevertheless, it is remarkable that such sparse information allows us to refine the coarse-grained spliceosome structure to a medium resolution.

The final example shows the power of DPS for 3D reconstruction from few projections and a subunit structure. This is a common scenario in structural biology where many partial structures have been determined and the challenge is to determine the full structure. To test this scenario, we model the 26S proteasome (PDB code 6fvt, 110541 heavy atoms). Historically, a huge part of the 26S proteasome, the 20S proteasome, was determined before the complete 26S structure could be elucidated by cryo-EM. In our tests, we use three projections and the structure of the 20S proteasome as input (Fig. 2E). The RMSD between the models obtained with DSP and the ground truth is $8.14 \pm 0.12$ Å (atomic structure) and $5.66 \pm 0.19$ Å (subsampled structure).

### 4.4 LIMITATIONS

A major limitation of the proposed method concerns its runtime. In each approximate DPS step with correction, we have to evaluate the gradient of the energy from our forward model twice. Overall, this means that we need $2 \times \#\text{timesteps} - 1$ network function evaluations and have to solve $(2 \times \#\text{timesteps} - 1) \times \#\text{observations}$ linear assignment problems to obtain a single 3D reconstruction. However, the time to reconstruct a 3D structure in the case of 6 input projections and 40 timesteps within a batch of 10 still takes $\approx 1.2$ min per sample on a A100 GPU in combination with an Intel Xeon Platinum 8360Y 2.40 GHz CPU.

## 5 CONCLUSION

We propose a Bayesian approach for 3D reconstruction from sparse measurements such as 2D projections, coarse-grained structures, and/or substructures, using diffusion models as priors. Diffusion models are capable of encoding rich prior information about 3D structures and enable us to reconstruct meaningful 3D models from very sparse input data via approximate diffusion posterior sampling. Diffusion priors can distill rich data sources and thereby complement existing regularization techniques whenever such training data are available. The goal of future research is to improve the resolution of the 3D reconstructions.

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
