# OpenReview forum: "Diffusion priors for Bayesian 3D reconstruction from incomplete measurements"
_ICLR.cc/2025/Conference — Submitted to ICLR 2025_

### Official Review · Reviewer_MTJn · 2024-10-27

**Soundness:** 3
**Presentation:** 2
**Contribution:** 2
**Rating:** 5
**Confidence:** 3

**Summary:**

The paper proposes a strategy for using a diffusion model for point cloud reconstruction from partial observations via reconstruction guidance. These partial observations may come from one or multiple inverse problems with known forward models; even though the observations are quite sparse (e.g. a handful of projected point clouds, or a subunit of the full 3D point cloud, or a low-resolution point cloud) the 3D diffusion model prior constrains the reconstruction to lie in the space of reasonable structures. The proposed method is validated on 3D point clouds from ShapeNet and from the Protein Data Bank.

**Strengths:**

- The proposed strategy is quite general within the context of 3D point cloud recovery, in the sense that it can operate with very sparse and very diverse types of measurements (as long as the forward model is known for each measurement).
- The method summary and theoretical introduction to diffusion models may be helpful to readers who are new to the topic, though they are not a novel contribution.
- The paper clearly shows benefits of diffusion posterior sampling compared to maximum likelihood estimation from the measurements alone (without the diffusion prior), across several inverse problems and datasets using 3D point clouds. I particularly appreciate the empirical demonstration that the maximum likelihood solution may not be in the support of the prior, if the prior is multimodal. This is shown in the paper via lower energy scores for maximum likelihood reconstruction, while visual and other quality metrics are higher for the diffusion posterior sampling reconstruction.

**Weaknesses:**

I subdivide the weaknesses into weaknesses in the experimental evaluation and weaknesses in the exposition.

Experimental weaknesses
- Figure 1 shows “prominent example reconstructions that demonstrate the superior performance of DPS” (line 418). I would prefer to see randomly-chosen examples so that I can appreciate the average-case performance rather than potentially cherry-picked cases.
- Line 422 says that the structural models from DPS “show a lower degree of heterogeneity” than the model from maximum likelihood…where is this claim supported by experiments? The standard deviations reported in Table 1 tend to show higher variance for DPS compared to ML.
- The cryo-EM experiments in section 4.3 are said to be focused on the sparse-data regime because of interest in studying conformational heterogeneity. On the one hand, I completely agree that conformational heterogeneity is of great interest. On the other hand, it is not clear why the proposed method would be useful for recovering heterogeneous structures, or why we would need to focus on the sparse-supervision setting to study heterogeneity.
- The cryo-EM experiments are all at relatively low resolution, while the authors acknowledge that recovering high-resolution details is the primary challenge in practice. What is the motivation for studying the medium-resolution setting, and are there any barriers to using your method at higher resolution?
- A major challenge in cryo-EM reconstruction is the low SNR of the projection images, yet in the experiments here it looks like the images used for sparse supervision are not noisy. I would find it more convincing to use many noisy projections for supervision rather than few noiseless projections, as this would be closer to the experimental setting for cryo-EM.

Presentation weaknesses (roughly in order of appearance in the paper)
- In the text immediately following eq (1) and eq (2) there is a helpful interpretation of the equation in the Gaussian noise case. However, the text after equation 1 describes equivalence to “standard least-squares fitting” whereas after equation 2 the equivalence is to “regularized least-squares fitting”…why are these different?
- Some recent related works should be discussed; these are listed under the questions section below in roughly chronological order. The most recent ones are concurrent work so I do not expect them to have been included in the initial submission, but the older ones are prior work that should be discussed. In particular, the contribution claim that “Using diffusion priors has previously not been explored to solve the 3D reconstruction problem in cryo-EM” should be made more nuanced in light of these prior works, to distinguish more specifically what is novel in the proposed approach.
- You might consider merging sections 3.1 and 3.2 into section 2, so that there is a smooth flow between introducing standard diffusion methodology and then the specifics of how you apply it to 3D point cloud reconstruction. In particular, equation 8 in section 3.1 is a generalization of equation 4 in section 2, and equation 9 in section 3.2 is nearly identical to equation 7 in section 2.
- Table 2 is not referenced in the main text. I would suggest moving it to be inside the new section 2/3 and referencing it there.
- It would be great to include more line-level annotations in Algorithm 1, i.e. on lines 8, 9, and 10 individually rather than a single combined annotation on line 7.
- This is a very minor LaTeX bug, but throughout the paper the opening quotes are all backwards.
- Figure 2 caption says that “the second row shows all ten point clouds generated with DPS”…this does not seem to match what is actually in the second row of the figure, which would appear to be a single structure for each experiment.

**Questions:**

Please refer to some questions embedded in the weaknesses section. I also have some questions around how this work relates to other recent methods that use diffusion models in cryo-EM. It is understandable to not compare against these methods experimentally given their recency (though that would be great), but a brief discussion of similarities and differences would benefit the paper.
- Latent Space Diffusion Models of Cryo-EM Structures (https://neurips.cc/virtual/2022/61226), was at NeurIPS 2022. This paper considers a diffusion model in the latent space rather than 3D space, so it’s not directly comparable but might be worth including in the discussion of related work.
- Illuminating protein space with a programmable generative model (https://www.nature.com/articles/s41586-023-06728-8), was published in 2023 and bears similarity to the proposed approach (diffusion in 3D).
- Robust Single-Particle Cryo-EM Denoising and Restoration (https://ieeexplore.ieee.org/document/10447135?denied=), was at ICASSP 2024 and appeared on arXiv in January 2024. This paper does diffusion denoising in image/projection space rather than in 3D.
- Solving Inverse Problems in Protein Space Using Diffusion-Based Priors (https://arxiv.org/abs/2406.04239), preprint released in June 2024 so this might be considered concurrent, or at least very recent. I think this is the most similar to the proposed approach in that it considers several different protein-related inverse problems, and uses the diffusion model as a plug-n-play prior.
- DiffModeler (https://www.nature.com/articles/s41592-024-02479-0), preprint was posted in January 2024 but the published version was just released a week ago.

---

> ### Author Response · Authors · 2024-11-18
>
> ***Figure 1 shows “prominent example reconstructions that demonstrate the superior performance of DPS” (line 418). I would prefer to see randomly-chosen examples so that I can appreciate the average-case performance rather than potentially cherry-picked cases.***
>
> We picked the test cases by diversity and better likelihood value of the baseline approach. The shown examples are not cherry-picked.
>
> ***DPS “show a lower degree of heterogeneity”***
>
> We agree that this claim was misleading and will be removed.
>
> ***The cryo-EM experiments in section 4.3 are said to be focused on the sparse-data regime because of interest in studying conformational heterogeneity. On the one hand, I completely agree that conformational heterogeneity is of great interest. On the other hand, it is not clear why the proposed method would be useful for recovering heterogeneous structures, or why we would need to focus on the sparse-supervision setting to study heterogeneity.***
>
> The major challenge in dealing with conformational heterogeneity are continuous movements. In these cases, the assumption that the system adopts distinct conformational states breaks down, and 3D classification approaches run into problems. In the face of continuous conformational changes, we would ideally aim to reconstruct a 3D structure from a single 2D image. This is why we focus on studying sparse data scenarios.
>
> ***The cryo-EM experiments are all at relatively low resolution, while the authors acknowledge that recovering high-resolution details is the primary challenge in practice. What is the motivation for studying the medium-resolution setting, and are there any barriers to using your method at higher resolution?***
>
> We do not see a fundamental problem in scaling our approach to a higher resolution. Our tests are manly limited by the available hardware.
>
> ***A major challenge in cryo-EM reconstruction is the low SNR of the projection images, yet in the experiments here it looks like the images used for sparse supervision are not noisy. I would find it more convincing to use many noisy projections for supervision rather than few noiseless projections, as this would be closer to the experimental setting for cryo-EM.***
>
> Thank you for the suggestion. We agree that noise is an important problem in cryo-EM reconstruction. However, as explained above your main focus so far has been 3D reconstruction from very few projection data with the aim of characterizing conformational heterogeneity. Tests with noisy projection data at low SNRs will be an important future test case.
>
> ***...the text after equation 1 describes equivalence to “standard least-squares fitting” whereas after equation 2 the equivalence is to “regularized least-squares fitting”…why are these different?***
>
> In regularized least-squares fitting, the objective is the log posterior $\log p(\mathbf{x} | \mathbf{y})$ which consists (beside a constant) of the log-likelihood $\log p(\mathbf{y} | \mathbf{x})$ plus the log-prior $\log p(\mathbf{x})$, while the objective in least-squares fitting only optimizes the log-likelihood part.
>
> **Some recent related works should be discussed; these are listed under the questions section below in roughly chronological order. The most recent ones are concurrent work so I do not expect them to have been included in the initial submission, but the older ones are prior work that should be discussed. In particular, the contribution claim that “Using diffusion priors has previously not been explored to solve the 3D reconstruction problem in cryo-EM” should be made more nuanced in light of these prior works, to distinguish more specifically what is novel in the proposed approach.**
>
> Thank you for the list of concurrent work. We agree with your statement and will discuss the papers in the revised version of our paper.
>
> ***Figure 2 caption says that “the second row shows all ten point clouds generated with DPS”…this does not seem to match what is actually in the second row of the figure, which would appear to be a single structure for each experiment.***
>
> In the second row all ten point clouds with each 1024 points are combined into a single point cloud with now 10240 points. This way we can visualize the uncertainty of the posterior.
>
> ***Table 2 is not referenced in the main text. I would suggest moving it to be inside the new section 2/3 and referencing it there.***
>
> Table 2 is referenced in the main text at line 212.

---

> > ### Comment · Reviewer_MTJn · 2024-11-22
> >
> > Thanks for the responses, some of which addressed my questions. To clarify a few points:
> >
> > > value of sparse-data regime for heterogeneous reconstruction
> > It is true that proteins can move in a continuous way, rather than bouncing between discrete conformations, so that there may not be many projections with exactly the same conformation. However, rather than viewing this as a challenge of reconstructing 3D from one (or very few) views, I would expect better performance by modeling the conformation in a continuous way (e.g. as done in CryoDRGN). So I'm not sure this is a clear motivation for considering the sparse-view case on its own.
> >
> > > regularization
> > I understand what regularization is and how it is implemented. My question was about why in this work you would choose to use regularization in one context but not another.
> >
> > > Figure 2 caption
> > Thanks for clarifying...I would suggest explaining this in the caption, and ideally color-coding the points corresponding to each of the 10 point clouds so that we can visualize the uncertainty, which is otherwise obscured by combining all the points. Alternatively, the 10 point clouds could be shown separately in an appendix figure.
> >
> > >Table 2 reference
> > Thanks for pointing this out; the reference was split across 2 lines so I missed it in my initial reading--though my suggestions regarding paper structure still stand.

---

### Official Review · Reviewer_XDb5 · 2024-10-27

**Soundness:** 3
**Presentation:** 2
**Contribution:** 1
**Rating:** 3
**Confidence:** 4

**Summary:**

This paper proposes a 3D version of Diffusion Posterior Sampling (DPS) to solve 3D reconstruction problems given incomplete measurements. The method is applied to shape reconstruction and cryo-electron microscopy. For the task of shape reconstruction, it is compared to a maximum-likelihood-based approach which does not incorporate priors.

**Strengths:**

* The proposed method is sound, and it appears to be a viable solution for 3D reconstruction from cryo-EM images.
* The writing and figures are clear.

**Weaknesses:**

* The technical contribution is not substantial. The only change from regular DPS is to use a point transformer in the diffusion model. The use of EDM (Karras et al. 2022) heuristics for the noise schedule and sampling scheme is also not a substantial contribution.
* The only baseline provided is a maximum-likelihood approach that does not use deep priors. What about other deep learning-based baselines, such as CryoDRGN (Zhong et al. 2020)? Without competitive baselines, it’s hard to assess the quantitative/qualitative results.
* In lines 75-76, the authors claim that DPS “has not yet been investigated in the context of 3D data.” This statement is much too strong, as extending DPS to 3D data is obvious. I’m sure it’s been investigated outside of published results, and just as one published example, Score-based Data Assimilation (Rozet and Louppe 2023) can be seen as an application of DPS to time-dependent 2D (i.e., 3D) data.
* In line 170, the authors say that “the guarantee of exactness is not of practical relevance.” Again this statement is too strong, and I don’t see how exactness would not be of practical relevance, especially for a scientific task like 3D cryo-EM.
* Related to the previous point, calling this a “Bayesian” approach feels too strong (“Bayesian-motivated” or “Bayesian-informed” feels more appropriate). The DPS approximation of the time-dependent likelihood is extremely crude, leading to a posterior that’s very far from the true Bayesian posterior. Furthermore, the authors do not analyze the uncertainty of the estimated posterior beyond giving error bars in Tables 1 and 2.

References:

Zhong et al. “Reconstructing continuous distributions of 3D protein structure from cryo-EM images.” ICLR 2020.

Rozet and Louppe. “Score-based Data Assimilation.” NeurIPS 2023.

**Questions:**

I would appreciate comments from the authors regarding why other deep learning-based baselines were not included.

---

> ### Author Response · Authors · 2024-11-18
>
> ***The only baseline provided is a maximum-likelihood approach that does not use deep priors. What about other deep learning-based baselines, such as CryoDRGN (Zhong et al. 2020)? Without competitive baselines, it’s hard to assess the quantitative/qualitative results.***
>
> Thank you for the suggestion to include CryoDRGN in the paper. We agree that it is hard to assess our results. However it is not trivial to use CryoDRGN as our baseline. CryoDRGN learns a continuous distribution of the structure of a macromolecular complex from many noisy cryo-EM images (in the order of $10^4$ images and more) and not from few (i.e. three to five) projections (as in our case). Moreover, CryoDRGN does not incorporate knowledge about already known biomolecular complexes and is only trained on data for a single macromolecular complex. Therefore, CryoDRGN needs more data (a large number of images) acquired for the particular biomolecular complex of interest. It is also not possible to incorporate knowledge about already known subunits or coarse-grained structures into CryoDRGN. These key differences in the input data make the comparison to our method challenging. We also emphasize that the idea of our diffusion model is not to replace state-of-the-art cryo-EM reconstruction methods such as CryoDRGN, but to complement these methods with a diffusion prior for macromolecular complexes.
>
> ***In lines 75-76, the authors claim that DPS “has not yet been investigated in the context of 3D data.” This statement is much too strong, as extending DPS to 3D data is obvious. I’m sure it’s been investigated outside of published results, and just as one published example, Score-based Data Assimilation (Rozet and Louppe 2023) can be seen as an application of DPS to time-dependent 2D (i.e., 3D) data.***
>
> Thank you for the comment, we were not aware of the named publication. We will weaken our claim to: “has not yet been investigated in the context of 3D point cloud data.”
>
> ***In line 170, the authors say that “the guarantee of exactness is not of practical relevance.” Again this statement is too strong, and I don’t see how exactness would not be of practical relevance, especially for a scientific task like 3D cryo-EM***
>
> We agree that exactness is of relevance, however exactness is only achieved when the number of particles in SMC approaches infinity. Since the number of neural network evaluations increases with the number particles, we are bounded by our computational capacity. For example, one could also do exact diffusion posterior sampling via importance sampling by first drawing $N$ samples from the diffusion prior $\mathbf{x}_i \sim p_θ$ and by assigning a weight $w_i = p(\mathbf{x}_i| \mathbf{y})$ to each sample. We could then sample from the posterior by drawing from $\sum_i^N w_i \delta (\cdot - x_i)$. As $N \rightarrow \infty$, this approach would also result in exact diffusion posterior samples, however, this method is not of practical relevance in high dimensions where importance weights become more and more imbalanced. But we agree that, in general, guarantees of exactness are of theoretical interest.

---

> > ### Comment · Reviewer_XDb5 · 2024-11-26
> >
> > Thank you to the authors for their response. Like other reviewers, I am still concerned about the lack of baselines, and I don't see any fundamental obstacle to implementing CryoDRGN as a baseline (it would still be helpful to see its performance with few measurements). Since my concerns still stand, I will keep the original rating.

---

### Official Review · Reviewer_xKFP · 2024-10-27

**Soundness:** 2
**Presentation:** 2
**Contribution:** 2
**Rating:** 3
**Confidence:** 3

**Summary:**

The paper investigates the application of a specific diffusion model-based method, named diffusion posterior sampling (DPS), to the 3D reconstruction problem in the context of CryoEM. The paper carefully describes how DPS can be adapted to CryoEM, detailed by explanation and formulation of the likelihood term. In the experiments, multiple datasets were used to demonstrate the better reconstruction performance achieved by DPS over the traditional maximum likelihood method. Overall, the main contribution of the paper is the application of DPS to the CryoEM problem.

**Strengths:**

The paper is clearly written.

**Weaknesses:**

1. **Insufficient Background Review:** There has been some literature on using diffusion models for solving the 3D reconstruction problem in Cryo-EM (Please see below). However, none of these or other similar work was reviewed and discussed in the background. Hence, it is unclear how the proposed method is different from the existing work.
   - DiffModeler: large macromolecular structure modeling for cryo-EM maps using a diffusion model. Nature Method
   - Latent Space Diffusion Models of Cryo-EM Structures. NeurIPS

2. **Lack of Novelty:** It appears that the proposed method is an adaption of DPS for Cryo-EM without novel algorithmic insights. While the formulation of the likelihood term is interesting, current manuscript lacks enough technical details to explain how the likelihood gradient is computed because the likelihood itself is an optimization problem.

3. **Missing Baselines:** The current experiments only include the maximum likelihood-based method as baseline, which is quite classic and cannot represent the current state-of-the-art. Thus, it is challenging to properly evaluate the performance of the proposed method in the context of modern Cryo-EM approaches.

**Questions:**

1. The authors are advised to provide more details on the computation of the likelihood and its gradient.
2. More recent methods should be included as baselines, such as Cryo-DRGN
   - 1 CryoDRGN: reconstruction of heterogeneous cryo-EM structures using neural networks

---

> ### Author Response · Authors · 2024-11-18
>
> ***Difference to DiffModeler***
>
> DiffModeler is a tool for fitting 3D structures (at atomic resolution) predicted by AlphaFold into an already existing cryo-EM map. DiffModeler uses a conditional diffusion model (no DPS!) to detect if a voxel in the cryo-EM map is part of the backbone or not. DiffModeler is not a diffusion model to generate cryo-EM maps, but receives a cryo-EM map as input to assemble a full-atom 3D structure.
>
> ***Difference to "Latent Space Diffusion Models of Cryo-EM Structures"***
>
> In this work the authors propose a method to train a diffusion model for the conformational space of a single protein. In combination with the decoder it can sample conformations for the particular system it was trained on. The model is also called a "diffusion prior" but for a different reason. It is not a diffusion prior for biomolecular complexes in general (like ours). It is only a prior on the conformational space of a single protein for which cryo-EM data were acquired.
>
> ***Difference to CryoDRGN***
>
> CryoDRGN learns a continuous distribution of the structure of a macromolecular complex from many noisy cryo-EM images (in the order of $10^4$ and more images) and not from very few projections (as in our case). Moreover, CryoDRGN does not incorporate the knowledge of already known biomolecular complexes and is only trained on data for a single macromolecular complex. Therefore, CryoDRGN needs more data (a large number of images) acquired for the particular biomolecular complex of interest. It is not possible to incorporate knowledge of already known subunits or coarse-grained structures into CryoDRGN. These key differences in the input data make the comparison to our method hard. We also emphasize that the idea of our diffusion model is not to replace state-of-the-art cryo-EM reconstruction methods such as CryoDRGN, but to complement these methods with a diffusion prior for macromolecular complexes.
>
> Thank you for suggesting the papers, we will mention them in our updated version.
>
> ***The authors are advised to provide more details on the computation of the likelihood and its gradient.***
>
> Distances between two point clouds always require (explicitly or implicitly) a point-to-point correspondence between points in two point clouds (e.g. 2D projection data and projected 3D model). To establish this correspondence, we first find the best correspondence between the two point clouds (by solving a linear assignment problem) which boils down to permuting the row order of the point cloud matrix. At this point no gradient calculation needs to be performed. After the optimal correspondence has been found, we can calculate the gradient with respect to the squared distances between the optimally assigned points.

---

### Official Review · Reviewer_4dGZ · 2024-11-04

**Soundness:** 3
**Presentation:** 3
**Contribution:** 2
**Rating:** 5
**Confidence:** 3

**Summary:**

This paper presents a framework for solving 3D inverse problems using diffusion models as a prior.  The diffusion-based posterior sampling (DPS) approach of Chung et al. (2023) is adopted.  Specifically, 3D structures are reconstructed from 2D projections using point-cloud based representations, adopting the point-cloud 3D diffusion model of Nichol et al. (2022).  The problem of cryo-electron microscopy is considered, demonstrating that the proposed approach allows 3D reconstructions from sparse, low-resolution and partial observations.

**Strengths:**

- A thorough but concise background on diffusion models in general, 3D diffusion models, and using diffusion models for posterior sampling is given, with fairly extensive references.
- Techniques are presented to further guide the diffusion prior, for example with a coarse 3D point cloud or by using information about subunits of the unknown 3D structure.

**Weaknesses:**

- Novelty is somewhat limited since Chung et al. (2023) is adopted for the posterior sampling framework and Nichol et al. (2022) is adopted for the 3D diffusion model.  The main contributions are in combining these for cryo-EM.
- For the ShapeNet problem, the benchmark considered (maximum liklihood) does not include any regularizing prior and it is therefore not suprising that performance is poor. It would be more informative to consider a method that includes a typical regularizing prior but that is not a generative model.
- For the cryo-EM problem, again a maximum likelihood alternative is considered as a benchmark.  There are no comparisons against existing standard methods.

**Questions:**

- How is a single estimated 3D shape recovered from approximate posterior samples?  In figures do you just show an example posterior sample or do you compute, for example, the mean?
- Do you use posterior samples to quantify uncertatinties?
- The benchmark maximum likelihood (ML) reconstructions do not impose any prior information, hence it is not surprising this approach performs poorly.  What type of regularizing prior would typically be considered for such problems?  An alternative prior might provide more meaningful results for comparison.
- For the cryo-EM problem, why is there an additional Truth row in Figure 2 based on a fitted mixture model?  Doesn't the PDB serve as the truth?

---

> ### Author Response · Authors · 2024-11-18
>
> ***How is a single estimated 3D shape recovered from approximate posterior samples?***
>
> The advantage of the Bayesian approach is that we do not obtain a single point estimate but a set of possible 3D structures reflecting the probability of observing them in the light of the observed measurements. There are multiple options to condense these samples into a single structure such as selecting the structure with highest posterior probability or computing the mean of the sampled structures. Here we analyze all ten sampled structures.
>
> ***In figures do you just show an example posterior sample or do you compute, for example, the mean?***
>
> In Figure 1 we showed one random samples from the approximate posterior per chosen task. In Figure 2 we combined per task (column) 10 random samples into a single point cloud. We have not used the sample mean to represent the posterior.
>
> ***The benchmark maximum likelihood (ML) reconstructions do not impose any prior information, hence it is not surprising this approach performs poorly. What type of regularizing prior would typically be considered for such problems?***
>
> We considered combining ML with a Gaussian prior, which penalizes points that are far from the origin. However, we found that choosing the regularization parameter (the inverse variance of the Gaussian) is not trivial and has to be done on a case-by-case basis. Moreover, even with an optimal regularization parameter the results are often not superior to unregularized ML. This does not come as a surprise, because for most test cases the likelihood is sufficiently strong to place the points at an appropriate distance from the origin.
>
> ***For the cryo-EM problem, why is there an additional Truth row in Figure 2 based on a fitted mixture model? Doesn't the PDB serve as the truth?***
>
> The structure that we label as "Truth" is the one obtained from the PDB file by clustering all heavy atoms. It is this structure that we used to generate the 2D point clouds and the partial structures ("subunits") that serve as input data. So ideally, a 3D reconstruction should recover this coarse-grained structure. The full-atom PDB structure is only shown for comparison and further validation.

---

> > ### Comment · Reviewer_4dGZ · 2024-11-25
> >
> > I thank the Authors for responding to my questions.  I have reviewed the other Reviewer comments and the Authors' responses.  My original rating of the paper stands.

---

### Meta-Review · Area_Chair_pAFS · 2024-12-20

**Metareview:**

Summary. This paper presents a framework for solving 3D inverse problems by adapting diffusion-based posterior sampling (DPS) approach. The focus is on the problem of cryo-electron microscopy, where the proposed approach allows 3D reconstructions from sparse, low-resolution and partial observations.

Strengths. The paper is well-written and the proposed method is sound.

Weaknesses. The technical contribution is limited as the paper mainly uses DPS method with a point transformer. The paper does not provide comparison with other methods. The experiments seem to be on the simulated examples. Cryo-EM methods suffer from low SNR, and the experiments in the paper largely ignore that aspect.

Missing.
The paper is missing technical and algorithmic novelty. Realistic experiments for 3D reconstruction could strengthen the paper, but the current experiments are not convincing from a practical standpoint. Comparison with existing methods is missing.

Justification.
Lack of technical novelty, lack of comparison with other methods, and experiments with noise-free simulated data are most important factors in my decision.

**Additional Comments On Reviewer Discussion:**

The paper had some discussion among authors and reviewers.

Reviewers mainly raised concerns about technical novelty (the method appears to be an adaptation of DPS method), lack of comparisons, and improvements in experiments.

Authors submitted their responses, but the reviewers were not convinced and maintain their scores leaning reject. AC agrees with reviewers.

---

### Decision · Program_Chairs · 2025-01-22

Reject